# A Unique Observation of a Patient with Vulto-van Silfhout-de Vries Syndrome

**DOI:** 10.3390/diagnostics12081887

**Published:** 2022-08-04

**Authors:** Natalia Bodunova, Maria Vorontsova, Igor Khatkov, Elena Baranova, Svetlana Bykova, Daniil Degterev, Maria Litvinova, Airat Bilyalov, Maria Makarova, Olesya Sagaydak, Anastasia Danishevich

**Affiliations:** 1The Loginov Moscow Clinical Scientific Center, 111123 Moscow, Russia; 2National Medical Research Center for Endocrinology, 117292 Moscow, Russia; 3LLC Evogen, 115191 Moscow, Russia; 4Russian Medical Academy of Continuous Professional Education, 125445 Moscow, Russia; 5Department of Medical Genetics, I.M. Sechenov First Moscow State Medical University (Sechenov University), 119048 Moscow, Russia; 6Institute of Fundamental Medicine and Biology, Kazan Federal University, 420008 Kazan, Russia

**Keywords:** VULTO-VAN syndrome, *DEAF1*, VSVS

## Abstract

Introduction: Vulto-van Silfhout-de Vries Syndrome (VSVS; OMIM#615828) is a rare hereditary disease associated with impaired intellectual development and speech, delayed psychomotor development, and behavioral anomalies, including autistic behavioral traits and poor eye contact. To date, 27 patients with VSVS have been reported in the literature. Materials and Methods: We describe a 23-year-old male patient with autism spectrum disorder (ASD) who was admitted to the gastroenterological hospital with signs of pseudomembranous colitis. ASD was first noted in the patient at the age of 2.5 years. Later, he developed epileptic seizures and important growth retardation. Prior to the hospitalization, chromosomal aberrations, Fragile X syndrome, and aminoacidopathies/aminoacidurias associated with ASD were excluded. Whole-genome sequencing (WGS) was prescribed to the patient at 23 years old. Results: The patient had a heterozygous carrier of “de novo” variant c.662C > T (p.S221L) in exon 4 of the *DEAF1* gene. c.662C > T had not been previously described in genomic databases. According to the ACMG criteria, this missense variant was considered to be pathogenic. VSVS was diagnosed in the patient. Conclusions: The phenotype of the patient is very similar to the data presented in the world literature. However, growth retardation and cachexia, which have not been described previously in the articles, are of interest.

## 1. Introduction

Vulto-van Silfhout-de Vries syndrome (VSVS; OMIM# 615828) is an orphan disease with an autosomal dominant type of inheritance, associated with impaired speech and intellectual development, delayed psychomotor development, and behavioral anomalies, including autistic behavioral traits and poor eye contact.

VSVS was first described by Vulto-van Silfhout (The Netherlands) in 2014 for four patients with delayed psychomotor and intellectual development with totally absent or impaired speech [1]. Currently, 27 cases of VSVS have been reviewed in the world literature. In addition to psychomotor abnormalities, most patients have additional nonspecific signs, such as hypotonia, gait disorders, seizures (that may be refractory), a high pain threshold, and sleep disorders [2].

VSVS occurs as a result of pathogenic germline mutations in the *DEAF1* gene, which encodes deformed epidermal autoregulatory factor-1 (*DEAF1*). *DEAF1* is involved in the regulation of multiple genes, showing high expressive activity in brain tissues, which explains its key role in the early development of the fetal central nervous system [3].

In most cases, mutations in the *DEAF1* gene are represented by “de novo” missense variants. The assessment of phenotypic–genotypic correlation showed no significant differences between heterozygous or homo- and compound-heterozygous mutation carriers, except for the presence of microcephaly in carriers of biallelic pathogenic variants in the *DEAF1* gene. The overall spectrum of detected variants of the *DEAF1* gene is described in Figure 1 (adapted from ‘De novo and biallelic *DEAF1* variants cause a phenotypic spectrum’/Maria J. Nabais Sá, MD, PhD. Genetics in medicine) [4].

The article describes a clinical case of a patient who had a heterozygous carrier of variant c.662C > T in the *DEAF1* gene responsible for the development of Vulto-van Sylfhout-de Vries syndrome. The mentioned variant of the *DEAF1* gene was not previously described either in the scientific literature or in any genomic database.

## 2. Clinical Case

The proband is a male patient (Russian/Ukrainian origin), 23-years-old, hospitalized to the State Budgetary Healthcare Institution—The Loginov Moscow Clinical Scientific Center—of Moscow Healthcare Department complaining of prominent fatigue, muscle weakness, weight loss, diarrhea (stool up to 2–3 times a day of mushy consistency and small amount, without pathological impurities), and edema of the lower extremities. The patient was accompanied by his mother, and could not establish personal or visual contact with the physician upon examination. According to his mother, the mentioned complaints appeared two months ago after receiving antibacterial therapy.

In terms of anamnesis vitae, the proband was born at 39 weeks of gestation from the first pregnancy complicated by development of the first trimester toxicosis. His weight and height at birth were 3200 g and 53 cm, respectively. The early motor development of the patient was normal.

In terms of anamnesis morbi, at the age of 2.5 years, mental developmental regression became evident, phrasal speech was missing, and the proband could say only individual sounds and syllables. According to the complaints mentioned, an autism spectrum disorder (ASD) was diagnosed. Epileptic seizures started at 5 years of age. Since then, seizures developed every 2–3 months. The proband received symptomatic treatment with anticonvulsants (valproic acid).

Growth retardation in the patient was first noted at 12 years of age. At 14 years old, due to severe stress and prominent appetite decrease, the patient started losing weight progressively. At 16 years old, the boy was hospitalized to the clinic to exclude an endocrine nature of his weight loss and growth retardation (weight 26.5 kg, SDS 6.68, height 143.5 sm, SDS-4.45). X-ray examination (and TW 20) showed that his bone age was 12.7 years old. The diagnosis on discharge was delayed physical and sexual development, i.e., cachexia (BMI 12.87 kg/m^2^ SDS 5.55). Brain MRI revealed no structural abnormality.

At 22 years old, the patient was first diagnosed with primary hypothyroidism, and therefore levothyroxine therapy was started. Adrenal as well as somatotropin insufficiency was excluded. There was an increase of the alkaline phosphatase level up to 147 IU/L (30–120), moderate hypoglycemia up to 3.7 mmol/L (4.9–5.1), and an increase of vitamin B12 up to 2179 pg/mL (189–833).

Throughout his life, numerous different genetic tests were conducted to identify the cause of the ASD in the proband. Numerical and large chromosomal aberrations such as micro-deletion and micro-duplication syndromes were excluded by karyotyping and DNA-microarray analysis (karyotype 46,XY; molecular karyotype arr(1–22)x2,(X,Y)x1). Molecular analysis of the abnormal methylation of the *FMR1* gene promoter region did not reveal any changes typical for Fragile X syndrome. Aminoacidopathies and aminoaciduria associated with ASD were also excluded by the tandem mass spectrometry of amino acids and organic acids in the blood.

The deterioration that led to the patient’s current hospitalization occurred 6 months prior to the admission, after the antibiotic therapy of a urinary tract infection followed by an increase in stool frequency up to 5–6 times a day. An abdominal CT showed a small bilateral hydrothorax, pronounced ascites, and an increased amount of gas in the intestinal loops. Upon a consultation with a gastroenterologist of the State B H I—The Loginov Moscow Clinical Scientific Center—of Moscow Healthcare Department, it was decided to hospitalize the patient for further investigation and treatment.

Upon admission, the patient’s condition was severe. The somatic status highlighted his short stature (143 cm), signs of hypogonadism, and cachexia (weight 31 kg). Consciousness was clear. The patient was sluggish, retarded, and demonstrated aggressive behavior and the active rejection of examination and medical interventions. No verbal contact was possible. The patient did not follow commands even to perform simple motor actions. According to the mother, the patient can write, but did not demonstrate any writing skills during the examination. Upon cranial nerve examination, the sense of smell was not evaluated, fields of vision were not limited, and there was full movement of the eyeballs: convergent strabismus due to the right eye, medium-sized pupils, S = D, normal eye slits, S = D, direct and cross photoreaction of the pupils preserved, satisfactory convergence, and no nystagmus was noted. Face surface sensitivity was normal. Masticatory muscle strength was 5/5 Medical Research Council Weakness Scale (MRC) points on both sides. The face was symmetrical and mobile. Hearing was not reduced. Swallowing was not impaired. Sternocleidomastoid and trapezius muscle strength was 5/5 MRC points on both sides. Muscle tone was slightly reduced. The upper and lower extremities’ strength was 4/5 MRC points on both sides. Tendon reflexes were of medium agility, S = D. No pathological symptoms were noted. The patient was not able to perform coordination tests, but did perform purposeful movements without an action tremor or discoordination. His mother moved him around in a wheelchair. The patient underwent routine electroencephalography (EEG), which did not record any typical epileptiform activity.

The patient was consulted by a gastroenterologist, a nutritionist and a neurologist. CT revealed hydrothorax on the left with the formation of compression atelectasis, signs of gastrostasis, a picture of inflammatory changes in the walls of the colon, total lesion in the active stage, and pronounced ascites. According to the laboratory data, there was a decrease in total protein down to 53 g/L (64–83), creatinine to 39 μmol/L (80–115), and hemoglobin to 8.2 g/dL (13.0–16.0); there was an increase in presepsin to 247 pg/mL (0–155). Clostridium Difficile was revealed (Toxin A and B positive).

Considering the presence of systemic pathology and ASD signs, consultation of clinical geneticist was performed. The patient’s phenotype was characterized by a short stature (143 cm), being underweight (31 kg), an impaired posture with kyphosis, a superficial location of the saphenous veins and marbling of the skin, multiple diffusely located nevi, freckles and small hyperpigmentation spots on the neck, an elongated face, hypertelorism of the eyes, a short nose, bilateral epicanthic folds, strabismus, an opened mouth, dry lips, attached earlobes, and impaired dermatoglyphics on the palms (Figure 2).

No clinical disorders similar to the disease of the proband were revealed in the family history (the patient has a healthy sibling, and the parents are not relatives to each other) (Figure 3).

Taking into account the negative results of the genetic tests performed, the geneticist recommended whole-genome sequencing.

The results showed a heterozygous variant chr11: 687913G > A (c.662C > T, p.S221L) in exon 4 of the *DEAF1* gene (NM_001293634.1). The identified variant is represented by a missense variant and leads to a substitution of serine for leucine at position 221 of the protein *DEAF1* (Figure 4A). A segregation analysis was carried out for the identified variant c.662C > T of the *DEAF1* gene in his mother, father, and sister. As a result of Sanger sequencing, the wild-type genotype (c.662C/C) was revealed in all of the relatives studied. (Figure 4B).

This variant had not been previously described in the gnomAD, 1000 G, ClinVar, ExAC, or Human Genome Mutation and Human Genome Variation. The substitution c.662C > T is localized in the highly conserved gene region (PM2), and the results of ten algorithms for the in silico prediction of pathogenicity (BayesDel_addAF, DANN, DEOGEN2, EIGEN, FATHMM-MKL, LIST-S2, M-CAP, MutationAssessor, MutationTaster and SIFT) confirm the pathogenic effect of the variant on the gene and its effect on protein splicing and functionality (PP3); additionally, more than 59.4% of variants (including missense mutations) in the SAND domain of the *DEAF1* gene are pathogenic (PM1). This codon previously described the missense variant chr11: 687914A > G (c.661T > C, p.Ser221Pro), annotated as likely pathogenic (PM5). Considering that the phenotype of the proband is highly specific for VSVS (PP4), and as a result of segregation analysis in the family, a “de novo” character of alteration in the patient (PP4) was established; according to the criteria of the American College of Medical Genetics, this nucleotide substitution can be considered as a pathogenic, clinically significant variant.

The final clinical diagnosis was: VSVS syndrome, epileptic encephalopathy, pseudomembranous colitis associated with ClDifficile (Toxin A and B positive), and metabolic disorders—hypoproteinemia, anasarca, ascites, and cachexia.

The patient’s treatment included antibacterial and symptomatic therapy to correct water-electrolyte disturbances due to long-term persistent pseudomembranous colitis.

The patient was discharged with some improvement in his condition under the supervision of specialists. Given the history of cachexia, with the identified genetic syndrome, and the associated long-term persistent infection (Cl. Difficile), the prognosis of the patient appears to be poor.

## 3. Discussion

ASD is an important medical and social problem. It is known that up to 30% of autism cases are associated with hereditary pathology, as both genetic and epigenetic disorders [5]. About 15% of ASD cases occur due to individual de novo mutations, while variations in the number of copies of chromosomal segment repeats (Copy Number Variation, CNV) additionally cause ASD in 5–10% of patients [6,7,8].

To date, 27 patients with VSVS have been reported in the world literature. In 81% of cases (22 out of 27 patients), the diagnosis was made before the age of 19, whereas for the patient described in this article, it happened at the age of 23 years old.

All of the patients had similar clinical manifestations; summarized data are shown in Table 1. The incidence of symptoms was calculated based on the data analysis from articles available on this issue in the world literature [1,4,5,9,10,11,12].

In most of these patients, additional nonspecific features were described, such as hypotonia and gait disturbance, a high pain threshold, and sleep disturbances [2].

The phenotype of the described patient is very similar to the data of the world literature. However, the growth retardation and cachexia, not previously described in the articles, are of interest.

## 4. Conclusions

ASD is an important medical and social problem characterized by the impaired neuropsychiatric development of the child with problems of patient interaction with society.

The article describes a patient with ASD and a polymorphic symptom complex, including developmental delay, which was the reason for performing whole-genome sequencing. As a result, the diagnosis of an extremely rare disease called VSVS syndrome was made. The disease turned out to be the result of a spontaneous mutation that occurred in one of the parents’ germ cells. The presented clinical case demonstrates the effectiveness of the use of mass parallel sequencing technology to find the cause and establish the prognosis of the disease.

## 5. Materials and Methods

All of the participants signed informed consent prior to genetic testing.

WGS was performed NGS on the MiSeq (Argonne, IL USA). The sequencing data results were analyzed using an automated algorithm, including the alignment of the reads to the reference sequence of the human genome (hg19), post-processing alignment, the identification of variants and the filtering of variants by sequencing quality, as well as the annotation of the identified variants according to all of the transcripts of each gene from the RefSeq database and using the predictive programs SIFT, PolyPhen2-HDIV, PolyPhen2-HVAR, MutationTaster, LRTPhyloP, and PhastCons.

The ACMG recommendations were used to annotate the identified variants. Samples from the 1000 Genomes, ESP6500, Exome Aggregation Consortium, and gnomAD projects were used to estimate the population frequencies of the identified variants. The OMIM database and worldwide literature data were used to assess the clinical relevance of the identified variants.

Genomic DNA was isolated from peripheral venous blood samples by the sorbent method (Sample-GS-Genetics, LLC DNA-Technology Research &Production, Moscow, Russia) according to the manufacturer’s protocol. DNA concentrations were measured with a Qubit 2.0 Fluorometer (Thermo Fisher Scientific, Waltham, MA, USA) using a Qubit ™ dsDNA High Sensitivity Assay Kit (Thermo Fisher Scientific). The isolated DNA was then amplified with a polymerase chain reaction (PCR) using oligonucleotide primers. Table 2 shows the Senger sequencing performed using the BigDyeTM Terminator v3.1 Cycle Sequencing Kit (Thermo Fister Scientific, Waltham, MA, USA) according to the manufacturer’s protocol. The detection of sequencing products was performed by Applied Biosystems 3500 (Thermofister Scientific, Waltham, MA, USA).

## Figures and Tables

**Figure 1 diagnostics-12-01887-f001:**
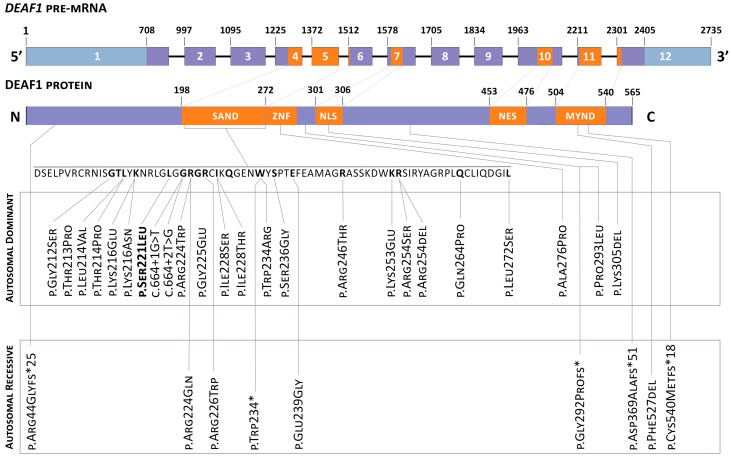
The structure of the *DEAF1* gene and the spectrum of gene variants detected earlier in patients with VSVS. * Termination of protein synthesis.

**Figure 2 diagnostics-12-01887-f002:**
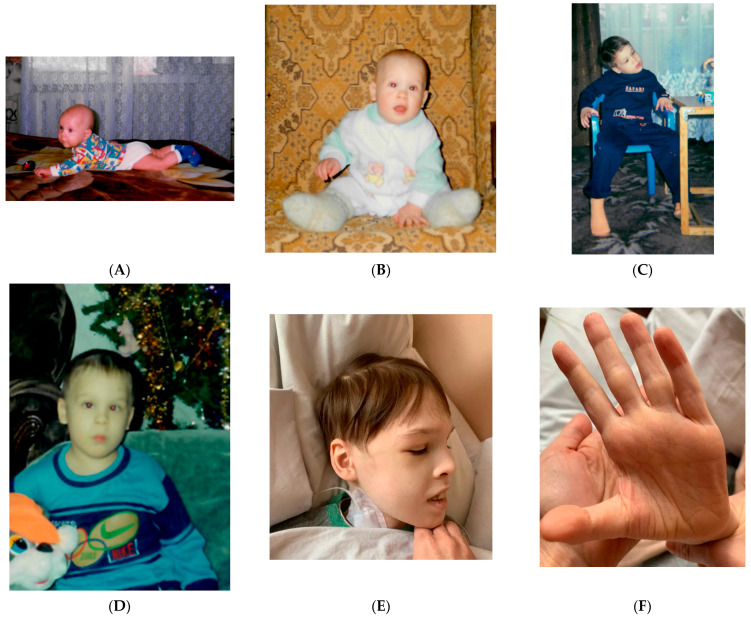
Phenotype of the patient with Vulto-Van Silfhout-De Vries Syndrome at different ages. (**A**) Patient at 5 months of age; (**B**) patient at 10 months of age; (**C**) patient at 4 years old; (**D**) patient at 6 years old, (**E**) patient at 23 years old, (**F**) dermatoglyphics of the palms of the patient at 23 years old.

**Figure 3 diagnostics-12-01887-f003:**
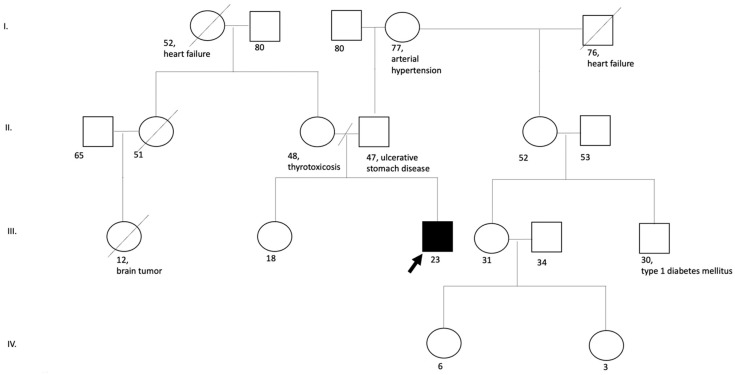
Pedigree of the patient with Vulto-Van Silfhout-De Vries Syndrome.

**Figure 4 diagnostics-12-01887-f004:**
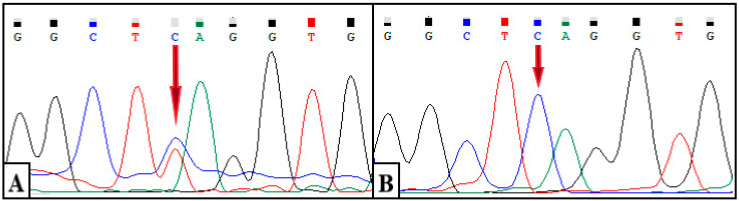
Electrophoregram of the nucleotide sequence of the *DEAF1* gene. (**A**) Substitution c.662C > T in the heterozygous state in the proband (indicated by the arrow). (**B**) The wild-type genotype at position c.662 in the sister, mother, and father of the proband (indicated by the arrow).

**Table 1 diagnostics-12-01887-t001:** The incidence of VSVS syndrome symptoms, according to the data described in the world literature.

Symptom	Incidence According to Literature, %	Presence of Symptoms in the Proband
Delayed psychomotor development	100	+
Gait disturbance (abnormal walking pattern)	95.7	+
Autism	96	+
Aggressiveness	77.8	+
Hypotonia	60.9	-
Dysmorphisms of the face	100	+
Poor speech/absence of speech	94.1	+
Seizures	79	+
Gastrointestinal abnormalities	81	+
Recurrent infections	70	+
Poor eye contact	75	+
Converging squint (strabismus)	not described in literature but can be noticed on the pictures of the patients from the published data (~25%)	+
Short stature	not described in literature	+
Cachexia	not described in literature	+

**Table 2 diagnostics-12-01887-t002:** Oligonucleotide primers of the polymerase chain reaction.

Oligonucleotides	F: 5′-GCCTCTCACTTCAAACACT-3′ R: 5′-CCACCACGCTCCACTAATTTT-3′			
PCR mix	Tris-HCl (pH = 8.8)	67 mM	25 µL	
	MgCl_2_	2.5 mM		
	Genomic DNA	4 ng		
	Primer F/R	5 pM		
	dNTP	10 mM		
	Taq polymerase	5 units		
PCR stages	Initialization denaturation	95 °C	5 min	34 cycles
	Denaturation	95 °C	30 s	
	Annealing	60–64 °C	30 s	
	Elongation	72 °C	15 s	
	Final elongation	72 °C	7 min	

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
