# Peer review of "A Unique Observation of a Patient with Vulto-van Silfhout-de Vries Syndrome"

_diagnostics, 2022, doi:10.3390/diagnostics12081887_

Round 1

Reviewer 1 Report

Bodunova report in high detail a case with Vulto-van Silfhout-de Vries syndrome (VSVS) with in addition to the characteristic features, growth retardation.

Comments:

1. The author mention hypotension as a feature for VSVS. They probably mean ‘hypotonia’ (line 46 and elsewehere, also in table 2). Please replace.

2. the photo at age 23 yrs are not very clear: a frontal photo would be better to assess the facial dysmorphology.

3 Table 2 needs major correction/revision!

a. The authors took the patients with de novo DEAF1 and the biallelic DEAF1 together. It is better to focus on the de novo cases in the SAND domain as these patients are defined as VSVS patients. So change all the percentages accordingly.

b. hypotension should be hypotonia.

c. ‘cutaneous syndactyly of the fingers’ should be ‘’ skin syndactyly in toes 2 and 3’. However this has not been observed/mentioned in the paper by Nabais Sá en thereby should be better removed.

d. ‘sacral fossa’should be sacral dimple’. This has not been reported anymore in the Nabais Sá paper and should be better removed.

Minor comments:

1. The disorder is rare but we  still have to find out whether it is ‘extremely’ as stated in the abstract. So please remove extremely (line17)

2. line 24 ‘he was excluded’ better use  ‘chromosomal aberrations…. were excluded’

3. Martin-bell is an old fashioned name for fragile X syndrome so please replace this throughout the text.

4. line 79: please write ‘Vulto-van Silfhout-de Vries syndrome’ correct here.

5. line 89: ‘The proband’ instead of ‘A proband’

6. line 100: 53cm instead of 53sm

7. line 102: play with toys

8. line 116: ‘SDS -4.45’ instead of ‘-SD -4.45’

9 unclear why line 120 is bold and put as a separate line.

10 line 130: remove ‘so’

11 figure 3: died/deceased  is indicated by on line from lower left to upper right (not a cross)

Author Response

Dear Reviewer 1, 

I am writing with regards to tell about, that we have corrected and uploaded our manuscript.  I would like to thank you for your Review Report.  All your comments were fixed.

I look forward to hearing from you.

Yours sincerely,

Dr. Airat Bilyalov. 

Reviewer 2 Report

Manuscript ID: diagnostics-1817696

Type of manuscript: Case Report

Title: A Unique Observation of A Patient With Vulto-Van Silfhout-De Vries Syndrome.

Comments to authors :

·       This is a well written manuscript relating a new Observation of a Russian patient with Vulto-Van Silfhout-De Vries Syndrome. This is an interesting report adding to previous reported patients with VSVS.

·       There are some modifications that will make the manuscript more relevant and complete:

The authors should add a table comparing their phenotype (in this report) with features and molecular findings of the VSVS, with other reported patients. This table will really give more value to the article. This new table will replace the table 2 (line 277 page 7).

·       This table should be discussed with a paragraph to add in the discussion section.

·       Some minor corrections should be done in the manuscript:

- Line 19: To date, 27 patients with VSVS have been reported in the world literature. Please just say reported in the literature

- Line 23 page 1 : Later he developed epileptic seizers and prominent growth retardation. Please change the sentence and say: Later he developed epileptic seizures and important growth retardation.

- Line 24 page 1: Martin Bell syndrome. Currently, this syndrome is named « X Fragile syndrome ». Please correct it in all the manuscript.

- Line 27 page 1: The patient appeared to be a heterozygous carrier of “de novo” variant c.662C>T (p.S221L). The authors should not say “appeared to be” but say “the patient had a heterozygous…”. Please correct in all the manuscript.

- Line 28 page 1 : in ex 4 of the DEAF1 gene. The authors should write « exon » instaed of « ex ».

- Line 186-187 : The patient's phenotype was characterized by short stature, underweight. Please give more information, the exact stature and weight measurements.

- Pedigree page 6 : what does mean the cross on the persons I1, I5, II2 and III1. If they are dead it is just a line on the person to specify that the person is dead. There is no cross in the international nomenclature of pedigree.

- Be sure that all the names of gene are in italic

Author Response

Dear Reviewer 2, 

I am writing with regards to tell about, that we have corrected and uploaded our manuscript.  I would like to thank you for your Review Report.  All your comments were fixed.

I look forward to hearing from you.

Yours sincerely,

Dr. Airat Bilyalov. 

Reviewer 3 Report

The authors present a 23-year-old male patient with VSVS due to DEAF1 gene variant. This syndrome is rare, therefore I think that description of clinical manifestation is valuable.

Major point:

As described by authors, the phenotype of this patient is very similar to the reported cases, and growth retardation and cachexia is  the only new observed points. Nonetheless, this manuscript is too long, and exceeds manuscript submission limit. The authors should shorten the manuscript to about 1,500 words.

Author Response

Dear Reviewer 3, 

Please kindly check attachment.

Yours sincerely

Dr. Airat Bilyalov. 

Round 2

Reviewer 3 Report

The authors revised the indicated points appropriately.